# Effects of Scanning Strategy on the Microstructure and Mechanical Properties of Sc-Zr-Modified Al–Mg Alloy Manufactured by Laser Powder Bed Fusion

**Yusufu Ekubaru [1,2], Ozkan Gokcekaya [1,2] and Takayoshi Nakano [1,2,\*]**

1    Division of Materials and Manufacturing Science, Graduate School of Engineering, Osaka University, 2-1 Yamadaoka, Suita 565-0871, Osaka, Japan

2    Anisotropic Design & Additive Manufacturing Research Center, Osaka University, 2-1 Yamadaoka, Suita 565-0871, Osaka, Japan

\*    Correspondence: nakano@mat.eng.osaka-u.ac.jp

**Abstract:** Laser powder bed fusion (LPBF)-manufactured Sc-Zr-modified Al–Mg alloy (Scalmalloy) has a bimodal microstructure comprising coarse grains (CGs) in the hot melt pool area and ultrafine grains (UFGs) along the melt pool boundaries (MPBs). Owing to these microstructural features, an increase in the MPBs can increase the UFGs, leading to enhanced mechanical properties. However, the effects of the LPBF process parameters, especially the laser scan strategy, on the microstructure and mechanical properties of Scalmalloy are still unclear. Here, a comparative study was conducted between X- and XY-mode laser scan strategies, with the same volumetric energy, based on the melt pool configuration, grain size distribution, and precipitation behaviors. The X-scan exhibited mechanical properties superior to those exhibited by the XY-scan, attributed to the higher volume fraction (VF) of UFGs. An increase in the VF of UFGs was observed, from 46% for the XY-scan to 56% for the X-scan, owing to an increase in MPBs. Consequently, the tensile strength of the X-scan was higher than that of the XY-scan. The maximum yield strength (271.5 ± 2.7 MPa) was obtained for the X-scan strategy, which was approximately twice that obtained for casting. The results of this study demonstrate that the microstructure and mechanical properties of Scalmalloy can be successfully tuned by a laser scanning strategy.

**Keywords:** laser powder bed fusion; Scalmalloy; ultrafine grain; precipitation; melt pool

## 1. Introduction

Laser powder bed fusion (LPBF)-based additive manufacturing (AM) is an innovative technology that fabricates a component layer-by-layer using a high-energy laser to selectively melt fine metal powders. LPBF is capable of fabricating almost fully dense, complex geometric parts with various advantages, such as a short time-to-market, high resolution, and considerable design freedom. LPBF has been attracting significant attention in the aerospace and automotive industries for the realization of lightweight isotropic components using high-performance Al alloys [1]. Studies on Al alloys and LPBF have demonstrated the manufacturability and performance improvements of Al-Si eutectic alloys such as AlSi12 [2,3], AlSi7Mg0.3 [4,5], AlSi10Mg [6], and Sc- and/or Zr-modified Al–Mg-based alloys (typically referred to as Scalmalloy) [7–13].

Al–Si eutectic alloys exhibit good processability in LPBF owing to their excellent weldability [1]; however, they exhibit anisotropic mechanical properties owing to the domination of columnar grains growing in the building direction, which forms under an extremely high thermal gradient [4–6].

Conversely, Scalmalloy, with a unique bimodal microstructure, exhibits superior mechanical properties that are almost isotropic with remarkable processability in LPBF. Consequently, Scalmalloy has garnered more attention than Al–Si eutectic alloys, and

studies focusing on improving the microstructure and mechanical properties of Scalmalloy have been conducted extensively [13–18].

Microstructural studies have revealed that Scalmalloy possesses a bimodal microstructure comprising coarse grains (CGs) oriented toward the melt pool center along the thermal gradient inside the melt pool, as well as ultrafine grains (UFGs) randomly oriented along the melt pool boundaries (MPBs) [13–18]. The CG region in the melt pool body is a higher-temperature region (>800 °C) where $Al_3$(Sc, Zr) precipitation can be dissolved, which can result in the formation of more CG structures with a lower density of precipitation [8]. However, the UFG region in MPBs is a lower-temperature region (<800 °C) where a considerably high density of $Al_3$(Sc, Zr) precipitates can form, which can lead to the formation of more UFG structures [8]. Furthermore, $Al_3$(Sc, Zr) precipitates form during both LPBF solidification and in situ aging treatment [19]. The precipitates formed in the MPBs during LPBF solidification play a critical role in the formation of the UFG structures. Owing to the aforementioned microstructural features of Scalmalloy, increasing the MPBs promotes precipitate formation and increases the UFGs.

Studies on mechanical property improvement have focused mainly on increasing the grain boundaries and precipitate strengthening by increasing the UFGs and precipitates, respectively, via various approaches [10,13,20]. Spierings et al. [10] demonstrated that a lower laser scan speed (*v*) promoted precipitate formation, leading to an increase in the UFGs and improved mechanical properties; this was attributed to the lower *v* decreasing the cooling rates and increasing the intrinsic heat treatment. Yang et al. [20] reported that the equiaxed grains increased as the platform temperature increased from 35 to 200 °C, and highlighted that the decreasing thermal gradient promotes precipitate formation. Further, Ekubaru et al. [13] demonstrated that a small laser hatch spacing (*d*) promotes precipitate formation and increases the volume fraction (VF) of UFGs owing to the promotion of track–track MPB formation where precipitates and UFG nucleation occur.

Hence, precipitates promote UFG formation, and precipitation is sensitive to changes in temperature and time, i.e., cooling rates associated with laser parameters; therefore, the optimization of laser parameters including laser scan strategies is necessary. To date, most research has focused on microstructural features and laser parameter optimization; however, these studies did not consider scanning strategy optimization. The scanning strategy is the geometrical pattern followed by the energy beam, which is an important factor that significantly influences the thermal history of each layer during LPBF and contributes to the modification of the microstructure, crystallographic texture, residual stress, and resultant mechanical properties [21–27].

The effects of scan strategy on the microstructure and mechanical properties of Scalmalloy have not been sufficiently studied to date. Thus, in this study, two types of scanning strategies, namely, X- and XY-, were adopted; further, the effects of different scanning strategies on the MPBs and the microstructural properties (e.g., porosity, grain structure, and crystallographic texture) and mechanical properties of Scalmalloy were studied.

This study demonstrates the feasibility of grain refinement by changing the scanning strategy, which leads to improvement in the mechanical properties of LPBF-manufactured Scalmalloy. Furthermore, the results of this study provide a basis for enhancing the mechanical properties, which in turn enhance the UFG, by controlling the morphology of MPBs.

## 2. Materials and Methods

### 2.1. Scalmalloy Fabrication via LPBF

The Scalmalloy powder used in this study (SCALMA40B5) was procured from Toyo Aluminium K. K. (Tokyo, Japan), and its composition and morphology are summarized in Table 1 and illustrated in Figure 1a, respectively. For convenience, SCALMA40B5 is hereinafter referred to as Scalmalloy. The particle size distribution of Scalmalloy was determined using a Mastersizer 3000E particle size analyzer (Malvern Panalytical, Malvern, UK) (Figure 1b). The volume-weighted percentiles of Scalmalloy powder were D10 = 31 μm,

D50 = 44 μm, and D90 = 62 μm. The avalanche angle and surface fractals were analyzed using a revolution powder analyzer (Mercury) to assess the flowability and estimate the homogeneity of the powder bed. The avalanche angle and surface fractal of the SCALMA40B5 powder were found to be 40.4° and 1.94°, respectively. These values were similar to those of commercial Ti-6Al-4V spherical powder (EOS, M290, Munich, Germany) (39.2° and 1.73°, respectively) [21]. Hence, the quality of Scalmalloy powder is adequate for LPBF processing.

**Table 1.** Chemical composition of scalmalloy in weight percent.

| Powder | Al | Mg | Sc | Zr | Mn | Fe | Si | Ti |
|---|---|---|---|---|---|---|---|---|
| SCALMA40B5 | Bal. | 4.8 | 0.74 | 0.29 | 0.57 | 0.10 | 0.04 | 0.02 |

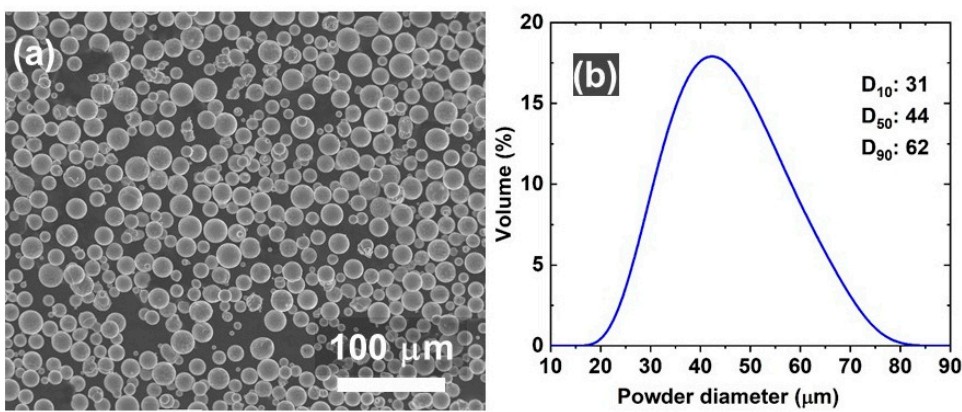

**Figure 1.** (**a**) Morphology of the Scalmalloy powders used in LPBF, and (**b**) powder size distribution.

An LPBF machine (EOS M290, Munich, Germany) with a 400 W Yb-fiber laser was used to prepare the Scalmalloy samples. Samples with dimensions of 5 mm × 5 mm × 15 mm along the *x*-, *y*-, and *z*-axes, respectively, were fabricated on an aluminum base plate using two different laser scanning strategies, namely, X- and XY-. The following parameter values were used: laser power ($P$) = 360 W, hatch space ($d$) = 0.1 mm, laser scan speed ($v$) = 1200 mm/s, and powder layer thickness ($h$) = 0.06 mm. The volumetric energy density ($E$) was calculated to be 50 J/mm$^3$. The calculated energy densities are listed in Table 2. The building stage was preheated to 300 K, and the building chamber was filled with high-purity argon gas to maintain the oxygen content below 100 ppm.

## *2.2. Microstructure Characterization and the Evolution of Mechanical Properties*

The samples were cut from the substrate by electrical discharge machining to investigate their microstructures. To observe the microstructure of the sample, the YZ plane was mechanically polished using emery paper (up to 4000 grade) and then chemically polished using colloidal silica to obtain a highly mirror-polished surface. Optical microscopy (OM) and electron backscatter diffraction (EBSD) measurements were conducted through field emission scanning electron microscopy (FE-SEM; JEOL JIB-4610F, Japan). The FE-SEM system equipped with EBSD detectors (Aztec HKL, Oxford Instruments, Abingdon, UK) was used at an accelerating voltage of 20 kV and a step interval of 0.5 μm. The data obtained were analyzed using analysis software (HKL Channel5, Oxford Instruments, Abingdon, UK) to obtain inverse pole figure maps and corresponding pole figures.

Tensile tests were conducted for the YZ plane, parallel to the build direction, in vacuum at room temperature and an initial strain rate of $1.67 \times 10^{-4}$ s$^{-1}$. The test was conducted thrice for each sample, and an average of the results was taken.

*2.3. Statistical Analysis*

The VF of UFGs and precipitates was calculated based on the EBSD band contrast (BC) and SEM backscattered electron (BSE) images using the Mathematica image-processing program developed by the authors. This program first filters the BC image and detects the black region, and afterward converts it to binary data. The black region in the processed image is then measured as the area% of the whole area, which is defined as a volume fraction. For the comparison of the yield stress (YS) data between the X- and XY- scan strategies, a two-tailed *t*-test was conducted, and $p < 0.05$ was considered to be statistically significant. The tensile test was conducted three times for each sample, and the results were averaged.

## 3. Results

Figure 2 shows the OM images of the LPBF-manufactured samples. The microstructures of the samples for the three planes were dense with no remarkable cracks but had small spherical pores; further, their OM densities reached 99.7%, which sufficiently met the industry-required density of >99.5% [23,28]. Spherical pores are generally formed during melting because of the trapping of the protection gas (Ar) or due to gases generated upon melting (through which metallic vapor is trapped at high cooling rates) [29]. Previous studies have shown that spherical pores with diameters ranging from 10 to 130 μm have no detrimental effect on the mechanical properties of the samples [28,30]. In this study, it was confirmed that the diameter of spherical pores was less than 100 μm; therefore, it can be inferred that the pores have no detrimental effect on the mechanical properties of the samples. These results indicate that changing the scan strategy does not affect the density of the sample. The dimension of specimens for the tensile test is shown in Figure 2c, and the sample thickness of the sample is around 1 mm.

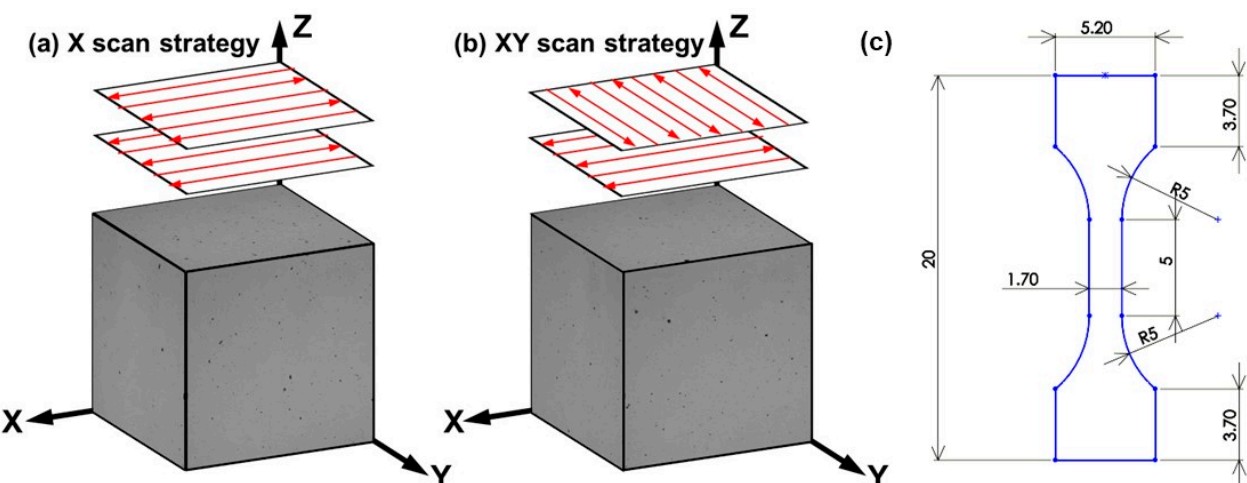

**Figure 2.** Three-dimensional OM images and the dimensions for the tensile test of the samples (**a**) X- and (**b**) XY-scan strategies; (**c**) the dimensions of the tensile test sample.

The microstructural features of the as-built Scalmalloy are schematically illustrated in Figure 3. Scalmalloy shows a bimodal microstructure, and the surrounding thick black bands correspond to MPBs comprising UFGs owing to a high concentration of $L1_2$ $Al_3$(Sc, Zr) precipitates. The CG columnar grains inside the hot melt pool area grow along the thermal gradient direction from the MPBs toward the top center of the melt pool. However, the UFGs are randomly oriented in the MPBs due to the presence of a large number of precipitates in the MPBs. This microstructural feature of Scalmalloy indicates that increasing the MPBs increases the UFGs. Furthermore, these MPBs can be tuned by changing the melt pool configuration via scan strategy alteration.

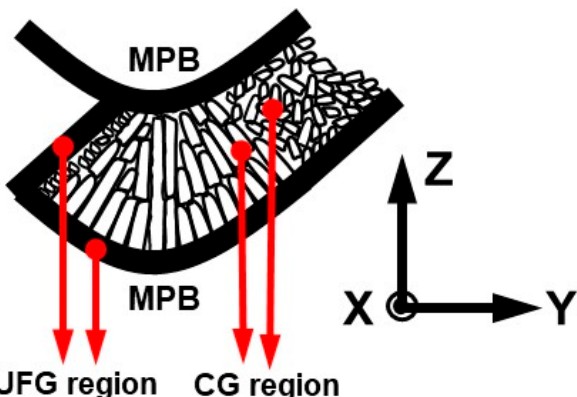

**Figure 3.** Schematic of the microstructural features of the as-built Scalmalloy.

Figure 4 shows three-dimensional EBSD grain orientation maps, extracted UFG regions with the VF of UFGs in every three planes, and the pole figures of the YZ plane of the X- and XY-scan samples. As shown in Figure 4a,b, the melt pools comprising black-band MPB and colorful melt pool bodies were clearly observed for the X- and XY-scan samples. The black-band MPBs correspond to the UFG with a size of <2 μm, and the remaining colorful regions correspond to the CG with a size of >2 μm [8,31]. Furthermore, the melt pool configurations were different in the X- and XY-scan samples. For the YZ plane, in the X-scan, all melt pools were arc-shaped configurations, whereas in the XY-scan, the melt pool configurations consisted of two kinds of melt pools, namely, arc- and band-like. For the XZ plane, in the X-scan, all melt pools were band-like with a gentle curve, whereas in the XY-scan, the melt pool configurations were the same as the YZ plane. On the other hand, for the XY plane, a discontinued melt pool track can be observed both for the X- and XY-scan.

These differences in the melt pool configuration naturally led to differences in MPBs, which affected the UFG VF. The calculated UFG VFs were different in the X- and XY- scans, as shown in Figure 4a',b'; the UFG VF was 57%, 56%, and 47% for YZ, XY, and XY planes, respectively, for the X-scan, and 46%, 46%, and 46% for YZ, XY, and XY planes, respectively, for the XY-scan. In addition to the melt pool configurations of the X- and XY-scans, the texture strength was different, as shown in Figure 4a',b'. In the X-scan sample, the CGs in the melt pool were inclined toward the center, and only a CG at the central position of the melt pool grew along the build direction with a (100) crystallographic orientation; this was expected because the rapid and directional solidification process promotes (100) preferential growth, as it is an easy growth direction for the FCC crystal structure [22]. However, in the XY-scan sample, numerous CGs tended to grow along the build direction with a (100) crystallographic orientation. Consequently, as shown in the pole figures in Figure 4a'',b'', both samples exhibited a (100)-crystallographic-oriented fiber texture, and the multiples of uniform distribution (MUD) within the pole figures were MUD = 2.04 for the X-scan and MUD = 2.84 for the XY-scan. The MUD number is one measure of the texture strength in an EBSD pole figure, with higher values indicating stronger alignment. and the XY-scan sample showed a slightly superior texture than the X-scan sample. This indicates that the X-scan tended to be more isotropic than the XY-scan.

Figure 5a,b display the BSE images of a region containing CG and UFG in the X- and XY-scan. The bright, small white dots in the BSE image represent the second-phase precipitates, which are concentrated in the UFG region in both samples. It was found that the precipitation VF in the X-scan (0.75%) was higher than that in the XY-scan (0.59%). These results indicate that the scan strategy significantly affected the microstructure behavior, such as the melt pool configuration, grain structure, and precipitates, which led to the difference in the UFG VF, owing mainly to the MPB configuration difference. This resulted in the X-scan demonstrating a microstructure superior to that of the XY-scan, including a high UFG VF, higher precipitation VF, and lower texture.

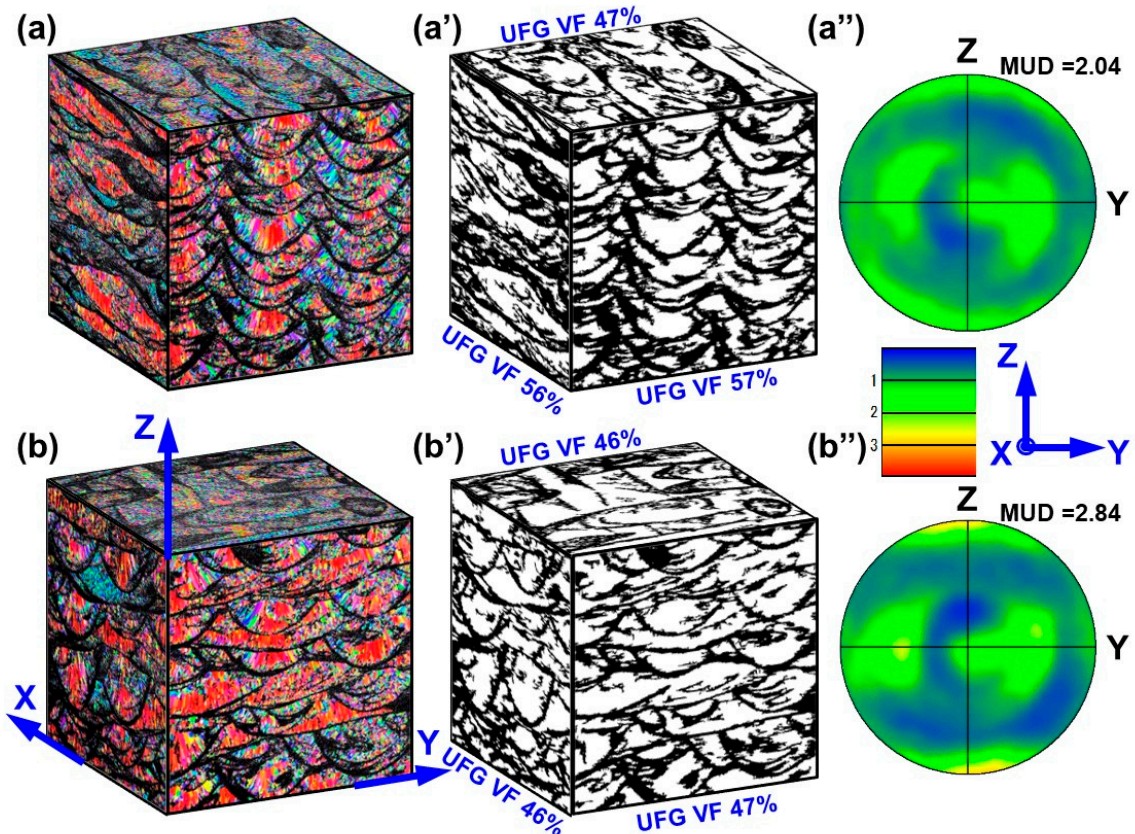

**Figure 4.** Three-dimensional (**a**,**b**) EBSD grain orientation maps and (**a'**,**b'**) extracted UFG regions with the VF of UFGs in all three planes; (**a″**,**b″**) the pole figures of the YZ plane of the X- and XY-scan samples.

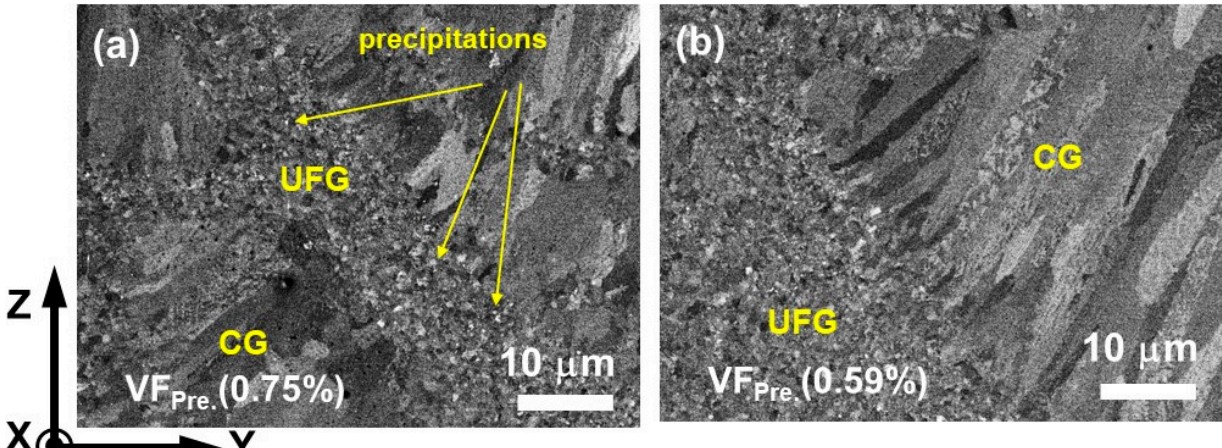

**Figure 5.** BSE images of (**a**) X- and (**b**) XY-samples in the YZ plane with the VF of precipitates.

To further investigate the composition of the precipitates, EDS line analysis was conducted on the X-scan sample, and the results are shown in Figure 6. The arc shape of the melt pools comprising the CG melt pool body and UFG band MPB is clearly visible (Figure 6a). Furthermore, using EDS line scan analysis, the bright, small white dots were identified to be $Al_3(Zr, Sc)$, as shown in Figure 6b; this result is in complete agreement with previously reported results [7,13,15,32]. The number density of the precipitates in the UFG band was considerably higher than that in the CG region, as shown in Figure 6a1,a2. These results prove that MPBs promote significant precipitation, which can be attributed to the

increased UFG formation. Therefore, it can be inferred that tuning MPBs by changing the scan strategy is an effective method for increasing the UFG VF.

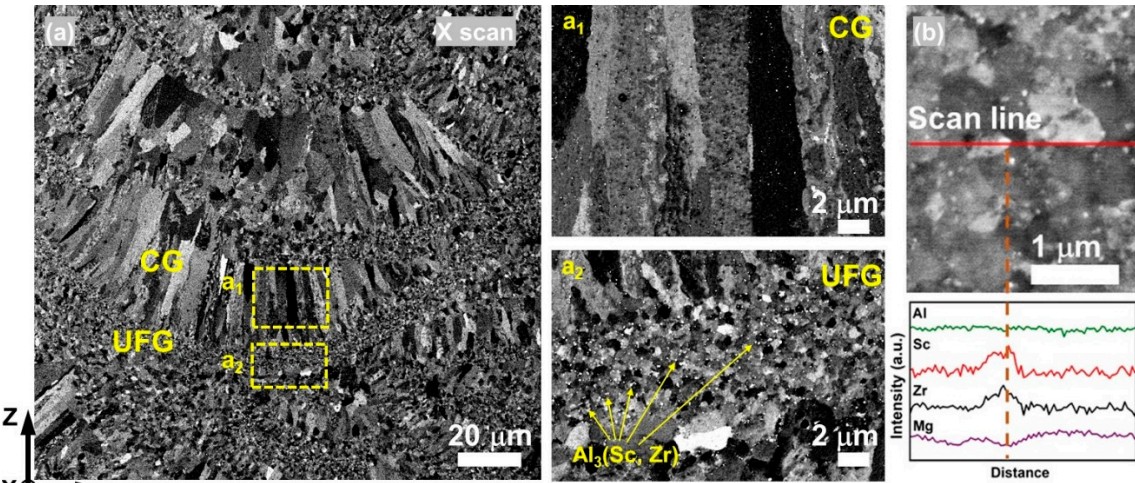

**Figure 6.** BSE images of (**a**) the YZ plane of the X-scan sample with the high-magnification images of (**a₁**) CG and (**a₂**) UFG regions, with corresponding (**b**) EDS element line analysis profiles of the precipitation.

Figure 7a shows the room-temperature stress–strain curves of the samples, and the values are listed in Table 2. The X-scan exhibited higher YS and UTS than those exhibited by the XY-scan, and both samples exhibited the Portevin–Le Chatelier serrated flow [9], which is typically observed in LPBF-fabricated Al–Mg-based alloys because of the dynamic interactions between Mg atoms and mobile dislocations during deformation [28]. As shown in Figure 7b, the YS of the X-scan significantly differed from that of the XY-scan, and the X-scan showed higher YS and UTS than those of the XY-scan owing to a higher UFG VF. The maximum YS was 271.5 ± 2.7 MPa for the X-scan sample, which was approximately two times higher than that of the cast alloy. As shown in Figure 7b, the elongation of the XY-scan was slightly higher than that of the X-scan because of the dominance of CG formation in the XY-scan. These tensile strength results corresponded to the microstructural features. The sample with a higher UFG VF showed better tensile properties, demonstrating that the mechanical properties of Scalmalloy can be improved by employing an X-scan strategy.

**Table 2.** Laser parameters, yield strength (YS), ultimate tensile strength (UTS), and elongation of the samples.

| Scan Strategy | d (mm) | P (W) | v (mm/s) | E (J/mm³) | YS (MPa) | UTS (MPa) | Elongation (%) |
|---|---|---|---|---|---|---|---|
| X-scan | 0.1 | 360 | 1200 | 50 | 271.5 ± 2.7 * | 352.2 ± 1.4 * | 29.4 ± 1.9 |
| XY-scan | 0.1 | 360 | 1200 | 50 | 261.3 ± 1.0 | 343.8 ± 0.6 | 30.5 ± 1.9 |
| Casting [33] | | | | | 131 ± 2 | 188 ± 2 | 3.7 ± 0.6 |

\* $p < 0.05$ significant difference.

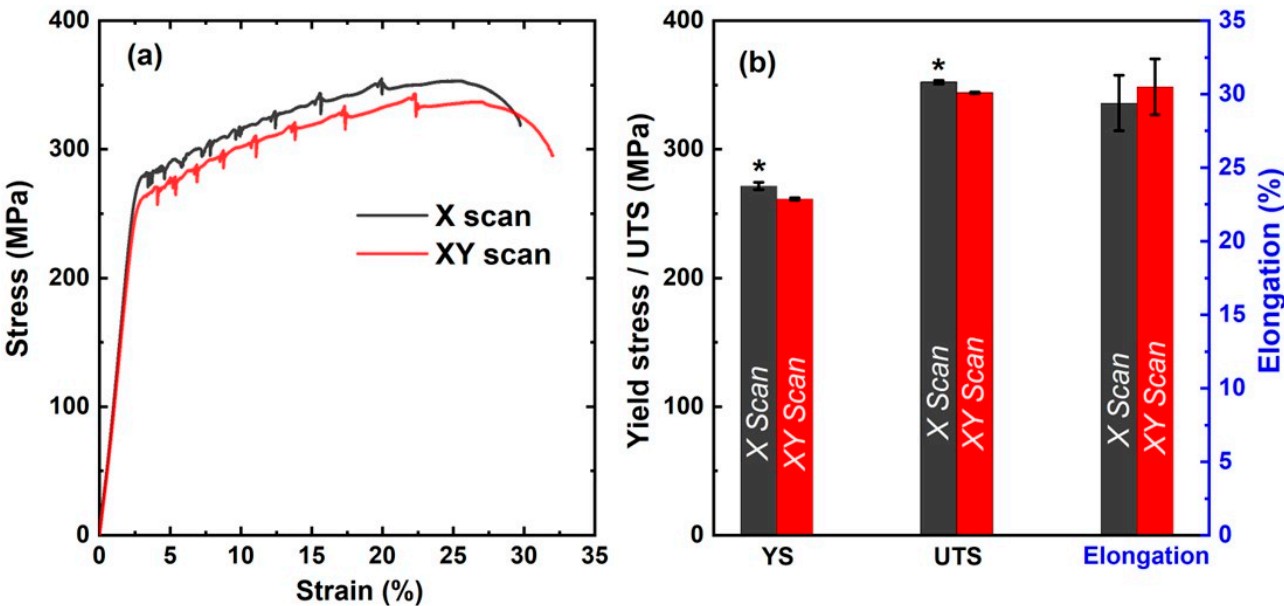

**Figure 7.** (**a**) Stress–strain curves, and the (**b**) yield strength (YS), ultimate tensile strength (UTS), and elongation of the samples (* $p < 0.05$ significant difference).

## 4. Discussion

### 4.1. Effect of Scan Strategy on the Microstructure

The effect of the scan strategy on the microstructure can be analyzed in terms of the melt pool configuration, i.e., MPBs, because the scan strategy significantly affects the melt pool configuration, which leads to a difference in the VF of UFGs and precipitates, as shown in Figures 4 and 5. As shown in Figures 3–5, in Scalmalloy, the UFGs are located in MPBs, which implies that increasing the MPBs increases the UFGs. Therefore, the MPB is an important factor that directly affects not only the microstructure, but also the mechanical properties of the Scalmalloy [13]. To distinguish between the behaviors of MPBs, such as the configuration and overlapping of X- and XY-scans, the MPBs were extracted from the BC images, shown in Figure 8.

As shown in Figure 8a',b', the MPBs of the X- and XY-scan samples consisted of approximately eight scan tracks, and the configuration and overlapping were clearly different. Compared with the XY-scan, in the X-scan, all eight scan tracks are arc shapes with high levels of overlapping, which increased in track–track MPBs. However, these behaviors are different in the XY-scan; the four X-scan tracks are arc shapes, and the four Y-scan tracks are band-like shapes. Furthermore, the overlapping is much lower, which can be attributed to the fact that the Y-scan cuts the overlapping parts of the X-scan. This leads to a decrease in track–track MPBs in the XY-scan. Owing to MPBs increased in the X-scan, it showed a higher UFG VF with respect to the XY-scan. In a previous study, which focused on increasing the UFG VF, the strategy employed was based on increasing the remelting zone volume by using a high *E* and decreasing the thermal gradient by increasing the baseplate temperature to obtain a highly equiaxed grain structure [20]. By contrast, this study used the intrinsic microstructural feature of Scalmalloy in which the UFG is located at MPBs; further, the adjustment of MPBs by changing the scan strategy was attempted. This method successfully demonstrated that the UFGs can be increased by using the X-scan strategy. Furthermore, these results are in agreement with those of a previous study [13], which reported that an increase in the MPBs increases the VF of UFGs. The results of this study also prove that an increase in the MPBs leads to an increase in the VF of UFGs.

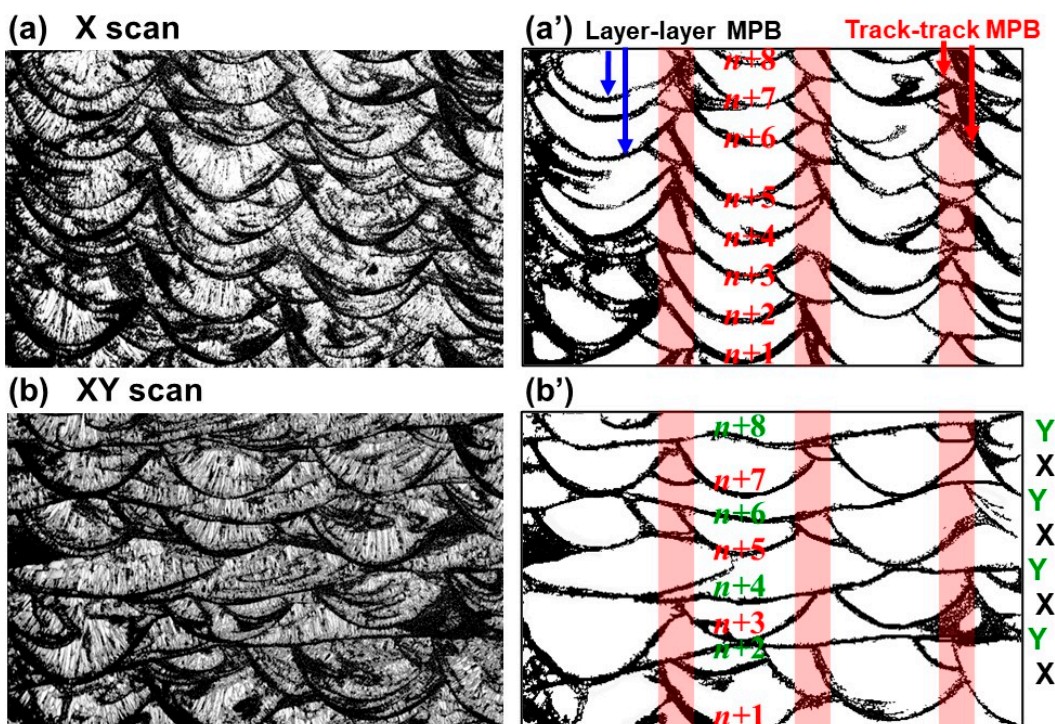

**Figure 8.** (**a**,**b**) BC images and (**a'**,**b'**) configurations of the extracted MPBs of X- and XY-scans.

*4.2. Effect of Scan Strategy on the Mechanical Properties*

Figures 4 and 5 show that the VF of UFGs and precipitates of the X-scan was higher than that of the XY-scan. Further, as shown in Figure 7, the X-scan sample has tensile properties superior to those of the XY-scan. The increase in grain boundary and precipitation strengthening can be attributed to the increase in the VF of UFGs and precipitates in the X-scan. The grain size distributions of these two samples for the YZ and XZ plane were further investigated, and the results are shown in Figure 9. Grains with a size of <2 μm are UFGs, while those with a size of >2 μm are CGs [8,31]. The results indicate that the UFG VFs are higher in the X-scan than in the XY-scan; however, the average grain sizes are similar in both the YZ and XZ planes, with values of 0.94 ± 0.16 and 0.93 ± 0.19 μm for the X- and XY-scans, respectively. By contrast, the number of fractions showed a similar tendency both in the YZ and XZ plane, and were higher in the X-scan up to 3 μm, while it decreased for values of X-scan greater than 3 μm. The average grain sizes of CG are 2.8 ± 0.2 and 3.2 ± 0.4 μm for both YZ and XZ planes for the X- and XY-scans, respectively. The results of these analyses indicate that the X-scan includes more fine grains for both the YZ and XZ planes than the XY-scan.

According to the Hall–Petch relationship [34], finer grains generate more grain boundaries per unit volume, which leads to an increase in the grain boundary strength.

$$\sigma_{GB} = V_{UFG}\, k\, d_1^{1/2} + V_{CG}\, k\, d_2^{1/2},$$

where $V_{UFG}$ and $V_{CG}$ are the VFs of UFGs and CGs, respectively; $d_1$ and $d_2$ correspond to the average grain sizes of the UFGs and CGs, respectively; and $k$ is the Hall–Petch slope with a value of 0.0.17 MPa/m$^{1/2}$ [13]. This equation indicates that $\sigma_{GB}$ is related to the grain size and VF; thus, increasing the UFG VF can increase $\sigma_{GB}$. Other strengthening mechanisms are attributed to precipitation strengthening ($\sigma_p$) because, as shown in Figure 4, the precipitate VF of the X-scan is higher than that of the XY-scan. Ekubaru et al. [13] demonstrated that the Orowan bypass mechanism is the main $\sigma_p$ in Scalmalloy. Therefore, from the aforementioned results, it can be concluded that UFGs and precipitates can be tailored by employing different scan strategies, which significantly affects the melt pool configuration.

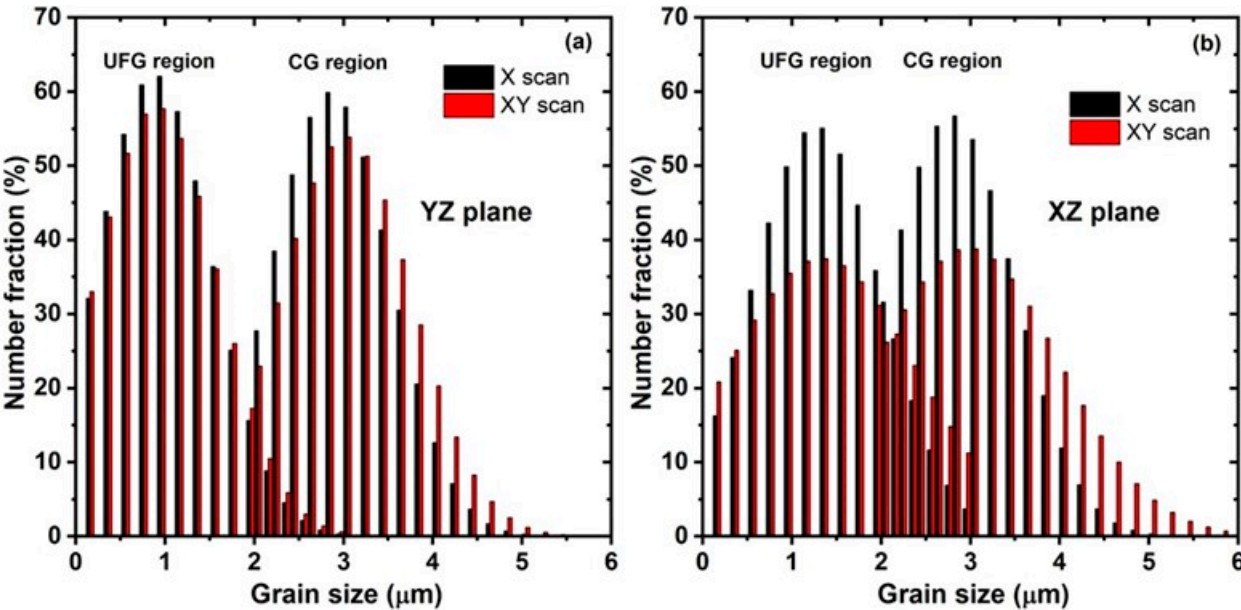

**Figure 9.** Grain size distributions of (**a**) the YZ and (**b**) the XZ plane for the X- and XY-scan samples with UFG and CG regions.

As shown in Figure 4, the X-scan exhibits a lower crystallographic texture and tends to be more isotropic than the XY-scan. Spierings et al. [31] reported an anisotropy in YS for the as-built Scalmalloy of <4%, and other studies showed lower values in heat-treatment conditions [35]; therefore, in this study, a tensile test was conducted only for the YZ plane.

The aforementioned results indicate that a combination of the X-scan strategy with adjustment of other laser parameters, such as the hatch space and scan speed, can facilitate further improvement in tensile properties by increasing the UFG VF.

## 5. Conclusions

In this study, the effects of scan strategy on the microstructure and mechanical properties of LPBF-fabricated Scalmalloy were studied. The following conclusions can be drawn:

(1) For the as-built Scalmalloy samples, a relative density of >99.5% was achieved by employing X- and XY-scan strategies and processing with laser parameters of $d$ = 0.1 mm, $P$ = 360 W, $v$ = 1200 mm/s, and $h$ = 0.06 mm.

(2) Detailed microstructural analyses revealed that the scan strategy significantly affected the microstructure and increased the VFs of UFGs and precipitates, which were greater for the X-scan strategy than for the XY-scan strategy owing to a higher amount of MPBs in the case of X-scan.

(3) Consequently, the tensile strength of the X-scan specimen was higher than that of the XY-scan specimen. The maximum YS (271.5 ± 2.7 MPa) was obtained for the X-scan strategy, which was approximately two times higher than that obtained for casting.

(4) Significant grain growth and differences in the precipitation behavior between the X- and XY-scan strategies were not observed.

In summary, the effects of scan strategies on the microstructure and mechanical properties of Scalmalloy with respect to the configuration of MPBs were demonstrated. The X-scan strategy increased the MPBs and promoted precipitation and UFG formation; consequently, the X-scan strategy produced higher tensile strength.

**Author Contributions:** Conceptualization, Y.E., O.G. and T.N.; investigation, Y.E. and O.G.; methodology, Y.E. and O.G.; supervision, O.G. and T.N.; validation, O.G.; writing—original draft, Y.E.; writing—review and editing, O.G. and T.N. All authors have read and agreed to the published version of the manuscript.

**Funding:** This work was supported by the Cross-Ministerial Strategic Innovation Promotion Program (SIP), Materials Integration for Revolutionary Design System of Structural Materials, Domain C1: "Development of Additive Manufacturing Process for Ni-Based Alloy" from the Japan Science and Technology Agency (JST) and CREST- Nanomechanics: Elucidation of macroscale mechanical properties based on understanding nanoscale dynamics for innovative mechanical materials (Grant Number: JPMJCR2194) from the Japan Science and Technology Agency (JST).

**Institutional Review Board Statement:** Not applicable.

**Informed Consent Statement:** Not applicable.

**Data Availability Statement:** Not applicable.

**Conflicts of Interest:** The authors declare that they have no known competing financial interests or personal relationships that could have influenced the work reported in this paper.

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
