# Peer review of "Effects of Scanning Strategy on the Microstructure and Mechanical Properties of Sc-Zr-Modified Al–Mg Alloy Manufactured by Laser Powder Bed Fusion"

_crystals, doi:10.3390/cryst12101348_

Round 1

Reviewer 1 Report

Overview

The Manuscript attempts to investigate the effect of scan strategy on a scalmalloy alloy using two different scan strategies. In the document it is claimed the change in scan strategies from XY to X increases the number of ultrafine grains observed in the microstructure which in turn increases the overall strength of the alloy. The manuscript is clear and concise and well structures. It clearly demonstrates that scan strategy will have a resulting effect on the microstructure of a system.

Comments

1.      In Line 135, perhaps be more specific about the image processing program. It would be interesting to see the methodology used in the image processing.

2.    In Lines 148-152, an argument is made that small pores from 10 to 130  have no detrimental effect on mechanical properties. Not only do neither of the papers cited [28,30] discuss Aluminum alloys at all, but there is not significant mention of porosity size in the 10 to 130  range effecting mechanical properties. [28] discussed reducing porosity but shows a clear correlation between more dense parts having improved properties. [28] also only investigates Tensile properties, stating mechanical properties is very deceiving. [30] does not talk about mechanical properties at all and only quantifies porosity size in Ti64.

3.      More information about how you preformed mechanical testing, on what size samples, and how many samples will be valuable information.

4.      The fundamental question is why does changing the scan strategy change the amount of UFG. In figure 4, it is apparent that cross sectioning has influenced the results of measurements. If the XY sample is cut in the XZ Plane, I would expect the microstructure to look largely the same, but if the X scan is cut in the XZ Plane, the number of ultrafine grains is likely reduced to only what is observed at the edge of weld tracks. The perpendicular cross sections to the XY and X scans should be investigated to make sure that there is truly a change in the amount of UFGs.

5.      Is it possible at all to quantify the microstructural overlap and generate a simple model that demonstrates the ratio of UFG and CG ratio. Something as simple as the Rosenthal weld model [1] would demonstrate the predicted thermal gradient and solidification velocity across a weld track in order to map observed microstructures to predicted thermal conditions.

6.      If such predictive studies existed, could a model steady state weld track be used to determine a “maximize” area of UFG region in a microstructure, taking the experimental data of X and XY scan strategies and allowing it to be used to predict the amount of UFG that exists in each scan strategy making a simple predictive tool.

[28] H.E. Sabzi, S. Maeng, X. Liang, M. Simonelli, N.T. Aboulkhair, P.E.J. Rivera-Díaz-del-Castillo, Controlling crack formation and 396 porosity in laser powder bed fusion: alloy design and process optimisation, Addit. Manuf. 2020, 34, 101360, 397 doi.org/10.1016/j.addma.2020.101360

[30] R. Cunningham, S.P. Narra, C. Montgomery, J. Beuth, A.D. Rollett, Synchrotron-based X-ray microtomography characterization 402 of the effect of processing variables on porosity formation in laser power-bed additive manufacturing of Ti-6Al-4V, JOM 2017, 403 69, 479-484, doi.org/10.1007/s11837-016-2234-1.

Author Response

Thank you for reviewing our manuscript and acknowledging the value of our findings. We greatly appreciate your time in providing detailed comments and recommendations. We have addressed these issues and revised the manuscript accordingly, as described in the attached file.

Reviewer 2 Report

The manuscript entitled "Effects of scanning strategy on the microstructure and mechanical properties of laser powder bed fusion manufactured Scalmalloy" was reviewed. In this manuscript, laser powder bed fusion (LPBF)–manufactured Scalmalloy was prepared and a comparative study was conducted between X- and XY-mode laser scan strategies. Although the author worked on a systematic study of the calculations, a number of questions are left open. Therefore, I would like to point out some problems in the manuscript.

1.      In Fig.5, according to the authors, the precipitates of Al3(Zr, Sc) can be observed and are important to the formation of UFG. However, there is lack of the structural characterization and confirmation of this precipitates. In my opinion, EDS analysis is not enough to confirm them, therefore, I suggest to supply XRD analysis to further prove them if it is possible.

2.      In the illustration of Fig.6, it maybe lacks the full name of YS.

3.      The title of Table2 includes the Vickers hardness of the samples, however, there is not Vickers hardness found in Table2.

4.      In line 223 on page 7, the word “diffed” should be “differed”.

5.      In line 251 on page 8, “can to attributed to” should be “can be attributed to”.

6.      In the references section, the format should be uniform. For example, the reference 1 lacks the Doi address.

Author Response

(The authors gave the same response as above.)

Round 2

Reviewer 1 Report

The previous comments have been addressed.